DOI: 10.1038/s41467-018-06918-3　　**OPEN**

# Molecular definition of group 1 innate lymphoid cells in the mouse uterus

Iva Filipovic [1,2,3], Laura Chiossone [4,10], Paola Vacca[5,6,7], Russell S. Hamilton [3], Tiziano Ingegnere [7], Jean-Marc Doisne[1,11], Delia A. Hawkes[1], Maria Cristina Mingari[5,6,8], Andrew M. Sharkey [3,9], Lorenzo Moretta [7] & Francesco Colucci[1,3]

Determining the function of uterine lymphocytes is challenging because of the dynamic changes in response to sex hormones and, during pregnancy, to the invading foetal trophoblast cells. Here we provide a genome-wide transcriptome atlas of mouse uterine group 1 innate lymphoid cells (ILCs) at mid-gestation. Tissue-resident Eomes$^+$CD49a$^+$ NK cells (trNK), which resemble human uterine NK cells, are most abundant during early pregnancy, and have gene signatures associated with TGF-β responses and interactions with trophoblast, epithelial, endothelial, smooth muscle cells, leucocytes and extracellular matrix. Conventional NK cells expand late in gestation and may engage in crosstalk with trNK cells involving IL-18 and IFN-γ. Eomes$^-$CD49a$^+$ ILC1s dominate before puberty, and specifically expand in second pregnancies when the expression of the memory cell marker CXCR6 is upregulated. These results identify trNK cells as the cellular hub of uterine group 1 ILCs, and mark CXCR6$^+$ ILC1s as potential memory cells of pregnancy.

[1] Department of Obstetrics and Gynaecology, University of Cambridge School of Clinical Medicine, NIHR Cambridge Biomedical Research Centre, Cambridge CB2 0SW, UK. [2] Department of Physiology, Development and Neuroscience, University of Cambridge, Cambridge CB2 3EG, UK. [3] Centre for Trophoblast Research, University of Cambridge, Cambridge CB2 3EG, UK. [4] G. Gaslini Institute, Genoa, 16147 Genoa, Italy. [5] Policlinico San Martino IRCCS per l'Oncologia, Genoa, 16132 Genova, Italy. [6] Department of Experimental Medicine (DIMES), University of Genoa, 16132 Genova, Italy. [7] Department of Immunology, IRCCS Bambino Gesù Children's Hospital, 00165 Rome, Italy. [8] Center of Excellence for Biomedical Research (CEBR), University of Genova, 16132 Genova, Italy. [9] Department of Pathology, University of Cambridge, Cambridge CB2 1QP, UK. [10]Present address: Innate Pharma Research Labs, Innate Pharma, 13009 Marseille, France. [11]Present address: Department of Immunology, Pasteur Institute, 75015 Paris, France. These authors contributed equally: Iva Filipovic, Laura Chiossone, Paola Vacca. Correspondence and requests for materials should be addressed to F.C. (email: fc287@medschl.cam.ac.uk)

Most innate lymphoid cells (ILCs) reside in tissues, where they integrate the local environment and its physiology. While group 2 and 3 ILCs are well characterised across tissues in humans and mice[1], the definition of group 1 (g1) ILCs is the most difficult due to their heterogeneity[2], as illustrated by human and murine liver g1 ILCs[3]. G1 ILCs include cytotoxic, conventional NK (cNK) cells and tissue-resident ILCs in liver, uterus, spleen, gut, salivary glands and thymus, which share with cNK cells expression of surface markers, transcription factor T-bet and production of IFN-γ. Little is known, however, about the physiological role of tissue g1 ILCs, whereas tissue ILC2s and ILC3s contribute to barrier integrity in lung and intestinal mucosa, promote tolerance of gut bacteria and regenerate lung epithelium upon viral infection[4]. G1 ILCs participate in early responses to infection through production of IFN-γ[5,6], however conversion of cNK cells into ILC1s under the influence of TGF-β undermines their anti-viral and anti-tumour responses[7,8]. Evidence also suggests g1 ILCs are involved in chronic inflammation in lung or intestine, where environmental cues drive ILC3s to convert into IFN-γ-producing ILC1s, which exacerbate pathology[9,10]. Thus, more information is available about tissue g1 ILCs in pathology than physiology[6].

Uterine ILCs contribute to optimal pregnancy outcome in mice[11–13] and g1 ILCs are the most abundant in both human and mouse uterus[14,15]. Among g1 ILCs, human uterine NK (uNK) cells maintain the integrity of endometrial arteries[16] and, during pregnancy, mediate key developmental processes and actively regulate placentation[17] and reviewed in ref. [18]. For example, they modulate trophoblast invasion, reshape uterine vasculature and promote foetal growth[17,19–21]. Genetic epidemiology studies have shown associations of pregnancy disorders with genetic variants of Killer-cell Immunoglobulin-like Receptors (KIRs) expressed on NK and some T cells and their variable HLA-C ligands[22,23]. Other functions have been suggested for uterine lymphocytes, including immunological tolerance[24], defence against pathogens[25,26], and roles in pregnancy complications such as miscarriage, although the evidence for this is controversial (reviewed in ref. [27]). Uterine ILC3s may also contribute to tissue physiology through production of IL-22, which maintains epithelial integrity[28]. A population of immature NK cells phenotypically overlaps with ILC3s, suggesting potential plasticity between uterine g1 ILCs and ILC3s[29]. Mouse uNK cells regulate uterine vascular adaptions to pregnancy[30] as well as foetal growth[31], but uterine g1 ILCs are heterogeneous[32] and could contribute to both physiology and pathology of reproduction[30,33].

Functional heterogeneity of uterine g1 ILCs may reflect division of labour, or result from the conversion of a subset into another under certain conditions determined by the stage of reproductive life orchestrated by sex hormones. Puberty, blastocyst implantation, placentation, parturition, and lactation are accompanied by remarkable tissue remodelling, which likely impacts on and is influenced by tissue lymphocytes. Additionally, ILC composition and function may be also marked by innate memory of pregnancy, which could contribute to the well-known better outcome of second pregnancies and their less frequent complications[34].

Determining the function of uterine cell types is challenging because of the changing nature of the organ and the limited access to human samples. Moreover, lack of knowledge on gene expression profiles of mouse uterine g1 ILC subsets precludes cell type-specific gene targeting approaches in mice. Modern immunology relies on systems biology to decode cell heterogeneity and ascribe functions to discrete subsets. Here we set out to begin to resolve the heterogeneity of g1 ILCs and provide a whole-genome transcriptome atlas of mouse uterine g1 ILCs.

We have previously characterised three uterine g1 ILCs[14], including Eomes+CD49a+ tissue-resident (tr)NK cells, which resemble human uNK cells, Eomes−CD49a+ ILC1s, which may be analogous to human uterine ILC1s;[13,15] and Eomes+CD49a− cNK cells, which are presumably circulating cells in both species. Here we determine their whole-genome transcriptional profile. The results show that trNK cells express genes that make them interact with most other cell types in the pregnant uterus and therefore, akin to human uNK cells, emerge as the central g1 ILC subset. cNK cells in the uterus may support the function of trNK cells by producing IFN-γ and responding to IL-18. ILC1s are most abundant before puberty, and CXCR6+ ILC1s specifically expand in the uterus, not in the liver, in second pregnancies, appearing as attractive candidates for memory cells. Determining the molecular identity and function of mouse uterine g1 ILCs may guide further work with human cells and generate opportunities for new treatments of patients with pregnancy complications.

## Results

**Dynamic distribution of uterine g1 ILCs.** We have previously defined three uterine g1 ILC subsets[13,14]. These subsets are all CD45+CD3−CD19−CD11b$^{low/-}$ NK1.1+ NKp46+ and their surface phenotype in comparison with that of liver subsets is shown in Fig. 1a. Here we set out to determine their distribution during key stages of mouse reproductive life. These were: just before puberty (3 weeks of age), during attainment of sexual maturity (5- and 8-week old), early in gestation (gd 5.5), at mid-gestation (gd 9.5), after placentation (gd13.5), late in gestation (gd17.5) and post-partum (days 1, 10 and 18). The relative abundance of different subsets at different stages was striking (Fig. 1b and Supplementary Fig. 1). Before the onset of puberty and exposure to sex hormones, Eomes−CD49a+ ILC1s are the most abundant. During sexual maturation, ILC1s decrease, while Eomes+CD49a+ trNKs increase. Upon mating, trNK cells are the most abundant in early pregnancy, on gd 5.5. Once the placenta has been established, at gd 13.5, trNKs decrease, while Eomes+CD49a− cNK cells become the most abundant. A similar landscape with cNK cells being the most abundant is observed both at days 1 and 18 post-partum, which mark respectively the beginning and the end stage of breast-feeding. Fertility is lower during lactation but not at the beginning or the end[35]. We observed marked differences in the distribution of g1 ILCs in the uterus of breastfeeding and non-breastfeeding females at 10 days post-partum, with breastfeeding females having increased trNK cells, similar to the distribution observed at mid-gestation (gd 9.5). These results show a dynamic g1 ILC subset distribution during reproductive lifes that likely reflects subset-specific functions.

**Genome-wide transcriptomes of uterine g1 ILCs.** Genome-wide transcriptional profiles of the three uterine g1 ILC subsets sorted from Eomes-GFP reporter mice at mid-gestation were analysed in multiple comparisons with the two, well-established liver g1 ILCs subsets. RNA was extracted from uterine and hepatic lineage negative CD45+CD11b$^{low/-}$NK1.1+NKp46+ cells sorted based on Eomes and CD49a expression as in Fig. 1a. This two-dimensional discrimination defines uterine and hepatic Eomes+CD49a− cNK cells, uterine and hepatic Eomes−CD49a+ ILC1s and, unique to the uterus, Eomes+CD49a+ trNK cells. Expression of CD49a therefore marks resident cells in both tissues[36]. Unbiased principal component analysis (PCA) showed shared transcriptional profiles between uterine ILC1s and trNK cells but highlighted unexpected differences between uterine and hepatic ILC1s. The PCA also confirmed similarities between uterine and hepatic cNK

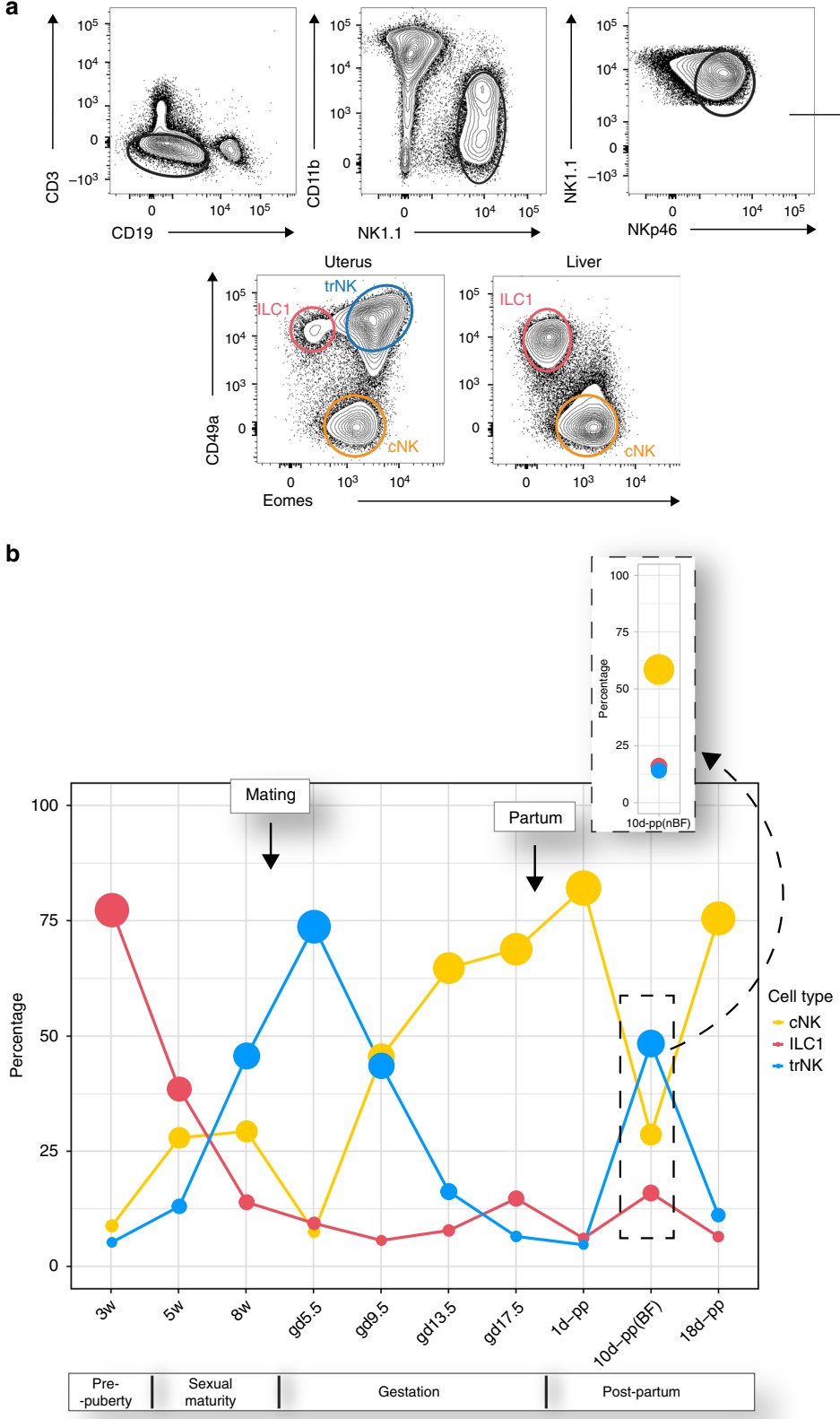

cells (Fig. 2a). Hierarchical clustering using the differentially expressed genes with a log2 fold change cutoff of 7.5 and *p*-value <0.01 confirmed that uterine and hepatic ILC1s cluster apart (Fig. 2b), despite phenotypic similarities (ref. [14] and Fig. 1a). The expression of *Tbx21* (T-bet), *Eomes*, *Ifng*, *Itga1* (CD49a), *Itga2*

(CD49b), *Klrb1c* (NK1.1), and *Ncr1* (NKp46) is consistent with the sorting strategy, though expression of *Tbx21*, *Ifng* and *Ncr1* is low in uterine ILC1s (Fig. 2c).

Surprisingly, genes associated with myeloid cells appeared among those defining the principal component 1 (PC1), which

**Fig. 1** Dynamic distribution of uterine group 1 innate lymphoid cells (g1 ILCs) during reproductive life. **a** Gating strategy for analysis and sorting by flow cytometry, with all freshly isolated g1 ILCs at mid-gestation (gd 9.5) gated on scatter and then defined as single live CD45$^+$CD3$^-$CD19$^-$CD11b$^{low/-}$NK1.1$^+$NKp46$^+$; ILC1 cells defined as CD49a$^+$Eomes$^-$; tissue-resident NK (trNK) cells defined as CD49a$^+$Eomes$^+$ and conventional NK (cNK) cells defined as CD49a$^-$Eomes$^+$. Each panel is representative of at least a hundred independent samples. **b** Landscape of uterine g1 ILCs during reproductive life; numbers in plot indicate mean percentage of each individual subset of g1 ILCs as gated within live CD45$^+$CD3$^-$CD19$^-$CD11b$^{low/-}$NK1.1$^+$NKp46$^+$ parent population. The size of the mean data points (filled circles) is proportional to the increasing percentage of the indicated subset. Inset shows the same time-point as the dashed rectangle below but in non-breastfeeding females. Data are representative of fifteen independent experiments with three individual animals for each time-point analysed. Mating for experiments (just after animals turn 8-weeks old) and partum (gd 19.5–21.5) time-points are indicated by the arrows on the plot. w week, d day, gd gestation day, pp post-partum, BF breastfeeding, nBF non-breastfeeding

explained 61% of the variance. These included *Clec7a* and *Clec4a3*, encoding Dectin-1 and DCIR3, respectively, *Adgre1* encoding F4/80, and *Mertk* encoding MERTK, a member of the Tyro-3/Axl/Mer (TAM) family of receptor tyrosine kinases (Supplementary Fig. 2A), highly expressed in CD49a$^+$ subsets compared to CD49a$^-$ cNK cells in both uterus and liver, thus defining transcriptomic differences between resident and circulating cells in both tissues. Flow cytometry on uterine subsets showed expression of F4/80 and CD86 on trNK cells, MERTK on both trNK and ILC1s and B220 on ILC1s, with trNK expressing intermediate B220 levels (Fig. 2d). The principal component 2 (PC2) explained 18.8% of the variance, with most genes significantly upregulated in liver ILC1s, with almost no expression in any of the other subsets, explaining in part why hepatic ILC1s do not cluster with uterine ILC1s (Supplementary Fig. 2B). These results mark the uterine resident CD49a$^+$ trNK and ILC1s as unique subsets.

**Core gene signatures of uterine g1 ILCs.** To compare g1 ILC transcriptomes between and among tissues, we ran 5 comparisons, the results of which are summarised in the UpSet plot, which also selects differentially expressed genes exclusively in the chosen comparison (Fig. 3a). The first comparison was between all liver g1 ILCs (ILC1 + cNK) versus all uterine g1 ILCs (cNK + trNK + ILC1). This comparison identified 265 differentially expressed genes (Fig. 3b). Gene Ontology (GO) analysis of this comparison for classification by biological process showed that the most highly enriched biological pathways in uterine cells relate to vascular endothelial growth factor (VEGF) signalling, including *Kdr*, encoding VEGF receptor 2 and *Pdgfra*, the alpha receptor for the platelet-derived growth factor (PDGF), but also oxygen/gas transport, including haemoglobin chains (*Hbb-Y, Hba-X, Hbb-BH1*) (Supplementary Fig. 3A and Fig. 3b). Other significantly enriched uterine pathways included response to hypoxia and decreased oxygen levels (*Ang, Hif3a*), extracellular matrix organisation (*Flrt2, Fbln1, Mmp9, Tgfbi*), as well as several pathways relating to wound healing and the regulation of blood vessel development. TGF-β signalling genes also distinguished uterine cells, with upregulated *Dab2, Tgfb2* and *Inhba* (Supplementary Fig. 3A and Fig. 3b). Supplementary Data 1 includes all differentially expressed genes and pathways highlighted by the 5 comparisons. *Trim2* is one of the genes differentially upregulated only in the uterus and *Itga3* encoding CD49c in the liver (Fig. 3b).

The second comparison was between liver CD49a$^+$ ILC1s and uterine CD49a$^+$ trNK and ILC1s, thus excluding cNK cells, and informing on tissue-specific profiles of CD49a$^+$ resident cells (Fig. 3c). The most highly upregulated genes in uterine CD49a$^+$ cells are *Gzmd, Gzme, Gzmf, Gzmg* encoding non-cytotoxic granzymes, likely involved in tissue remodelling. Indeed, other highly expressed genes regulate extracellular matrix organisation (*Flrt2*), collagen homeostasis (*Ctsb, Ctss*), wound healing (*Hmox1, Pdgfa, Plau*), and proliferation of epithelial cells (*Mmp12, Vegfa*)

(Fig. 3c and Supplementary Fig. 3B). Upregulated in uterine CD49a$^+$ cells are also genes involved in metabolic regulation of steroids (mainly progesterone metabolism—*Dhrs9* and *Srd5a1*), ketones (*Slc37a2*) and amines (*Paox*). Encoding the orphan monocarboxylate transporter MCT13[37], *Slc16a13* may be specifically upregulated in uterine CD49a$^+$ cells (Fig. 3c). Other upregulated genes in CD49a$^+$ uterine cells are genes regulating migration of both myeloid cells (*Ccl2, Ccl12, Spp1*) and lymphocytes (*Ccl7, Ccl8, Ccl17*). Osteopontin-encoding *Spp1* was reported to be upregulated in uterine NK cells[38] and we find that alymphoid *Rag2$^{-/-}$Il2rg$^{-/-}$* dams express less *Spp1* transcripts (Supplementary Fig. 3G). Two other growth-promoting factors, Pleiotropin-encoding *Ptn*, and Osteoglycin-encoded *Ogn* have also been implicated in NK-cell mediated foetal growth[39], however other cells produce these growth-promoting factors[40] and we found no significant difference in expression between wild-type and alymphoid uterus, thus excluding *Ptn* and *Ogn* as NK-cell and lymphocyte-specific factors driving foetal growth (Supplementary Fig. 3G). Expressing *Arg1, Arg2, Ecm1*, and *Tnfsf4*, uterine CD49a$^+$ cells may regulate type-2 immunity. Unexpected was the discovery of pathways regulating cellular import and phagocytosis, usually associated with antigen-presenting cells. The top enriched pathway in CD49a$^+$ liver cells relative to CD49a$^+$ uterine cells was negative regulation of antigen processing and presentation of peptide via MHC class II (Supplementary Fig. 3B). *H2-Oa* and *H2-Ob* (also identified previously in splenic CD127$^+$ ILC1s[2]) encode H2-O, an inhibitor of MHC class II (H2-DM)-mediated antigen-loading process. Moreover, liver CD49a$^+$ cells may regulate T-cell migration, with upregulated *Cxcr3* and *Ccr6*. Indeed, *Ccr6* may be exclusively expressed in liver ILC1s. Upregulated in liver cells are also *Cd27, Cd69* and *Cxcr6*, which encodes a CXCL16 receptor associated with 'memory' liver NK cells[41] (Fig. 3c).

The third comparison between the two CD49a$^+$ uterine subsets versus CD49a$^-$ uterine cNK cells informs on characteristics specific to uterine resident cells (Fig. 3d). Tissue resident cells highly express genes involved in induction of cell death (*Fasl, Hvcn1, Tnf*), response to both type I interferons and IFN-γ (including *Ccl7* and very highly, *Irf7*) and proteolysis, with nearly all cathepsins significantly upregulated, alongside *C3, Plau* and *Cpq* (Supplementary Fig. 3C). This suggests that uterine resident cells are involved in protein processing during tissue remodelling. Enriched pathways are chiefly unsaturated fatty acid biosynthesis (*Mgst1*) and lipid metabolism (*Apoe, Psap*) (Fig. 3d and Supplementary Fig. 3C). When analysed by flow cytometry, uterine Eomes$^+$ CD49a$^+$ trNK cells are highly granular[14], like their human counterpart. Consistent with this, the lysosomal transport (*CD68, Rab7b, Lamp1* and *Lamp2*) is enriched in tissue resident cells. The most upregulated genes in resident CD49a$^+$ cells are *Ccl7, Ccl2* and *Tlr1* (Fig. 3d). RT-PCR analysis confirmed higher expression of *Ccl2, Ccl7*, and *Ccl12* by uterine trNK (4.9, 9.7 and 25 folds, respectively) and ILC1s (1.4, 3 and 8.3 folds, respectively), compared to cNK cells. Tissue resident cells

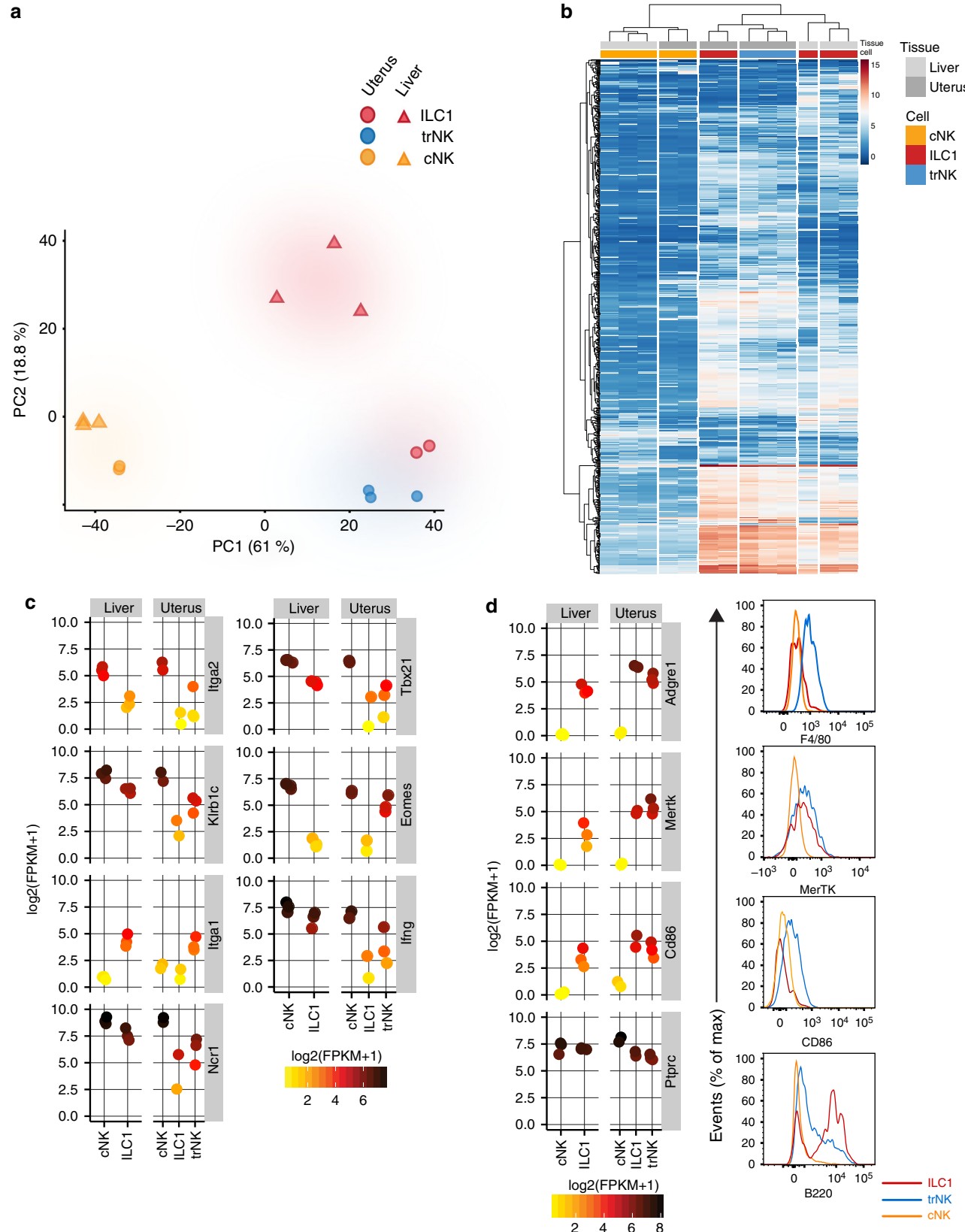

specifically upregulate the transcriptional repressor *Hic2*, as well as *Il6*, *Hint2* and *Clec4b1*. Highly upregulated exclusively in uterine cNK cells, as reported previously for splenic cNK cells[2], is *S1pr5* encoding sphingosine-1-phosphate receptor 5, which regulates NK-cell egress. One of the highest fold-change

upregulated genes in uterine cNK cells is *Muc4* encoding Mucin4, involved in epithelial cell integrity and homoeostasis.

The fourth comparison was between uterine trNK cell and cNK cells. The mutually exclusive expression of CD49a (*Itga1*) in trNK cells and CD49b (*Itga2*) in cNK cells reflected the cell-sorting

**Fig. 2** Genome-wide transcriptomes of uterine g1 ILCs at mid-gestation. **a** Unbiased principal component analysis (PCA)-based clustering of uterine ILC1s, trNK and cNK cells and of liver ILC1s and cNK cells. The 250 most variable genes and the two principal components were used for clustering and to describe the variance between the subsets. **b** Heat map of all significant differentially expressed genes with a log2fold change >7.5, selected from comparisons described in the text and presented in Fig. 3a. **c** Individual gene expression plots of genes encoding proteins used for sorting g1 ILCs and additional genes used for validation of the ILC and NK-lineages. Scale represents log2(FPKM + 1) transformed normalised reads. **d** Individual gene expression plots of genes encoding F4/80, MERTK, CD86 and B220 in uterine and liver g1 ILCs and flow cytometry validation of their protein products. Histograms show protein expression on three uterine g1 ILCs gated as gated in Fig. 1a (red line-ILC1s, blue line-trNK, orange line-cNK). FPKM fragments per kilobase of exon model per million reads mapped

strategy (Fig. 3e). About 900 pathways were detected in this comparison (Supplementary Fig. 3D), including those involved in responses to IFN-γ and IL-18. Upregulated in trNK cells are *Il18* and *Ifngr2*, while *Ifng*, *Il18r1* and *Il18rap* are upregulated in cNK cells. This suggests that trNK cells respond to IFN-γ and produce IL-18, while cNK cells respond to IL-18 and produce IFN-γ, (Fig. 3e). trNK cells may also respond to and produce IL-1β, through highly expressed *Il1r1*, *Ccl17*, *Ccl8* (cellular response to IL-1) and *Il1b*, *Nlrp1b*, *Nlrp3* (IL-1β production). Among the most highly enriched pathways in trNK cells were those related to cholesterol storage and homoeostasis, with genes such as *Apob*, *Lpl* and *Pparg* upregulated in trNK cells (Fig. 3e and Supplementary Fig. 3D). Interestingly, apoptotic cell clearance was among the enriched pathways, with *Lrp1*, *Axl* and *Mertk* highly upregulated in trNK compared to cNK cells. *Lrp1* encodes CD91, a receptor of many ligands and functions. In addition to roles in apoptosis, and more widely being a haemoglobin scavenger receptor, it also binds growth factors, including TGF-β. Therefore, uterine trNK and ILC1s may be highly responsive to TGF-β, through receptors such as CD91, Tgfbr1 and Tgfbr2. In addition, *Tgfbi*, encoding transforming growth factor-beta induced protein (TGFBIp, BIGH3), is also upregulated in trNK cells. UpSet pointed to *Slc16a1*, encoding the cell exporter of lactate MCT1, to be uniquely upregulated in trNK cells, emphasising the potential importance of anaerobic glycolysis for fuelling uterine CD49a⁺ cells[42]. Biglycan, encoded by *Bgn*, is a CD44 ligand that may be involved in the recruitment of circulating CD16⁻ NK cells into human endometrium[43] and is upregulated in trNK cells (Supplementary Fig. 3F). Regulators of cellular extravasation such as *Adam8*, *Ccl2* and *Ptafr* are upregulated in trNK, while *Itga4* and *Fam65b* are upregulated in cNK, suggesting migratory behaviour in both subsets, albeit through potentially different mechanisms (Supplementary Fig. 3F). *Ly86*, *Ly96*, *CD180*, *Sash1* are all significantly upregulated in trNK and part of highly enriched lipopolysaccharide-mediated signalling pathway. MyD88-dependent toll-like receptor (TLR) signalling is also specific to trNK cells, with *Tlr1*, *Tlr2*, *Tlr8*, and *Tlr9* highly upregulated (Supplementary Fig. 3F). Lysosome organisation pathway is enriched in trNK, just like complement and haemostasis regulation with *C3*, *F7*, *F10* and *Ptrpj*. Also, *C3ar1* and *C5ar1*, involved in complement receptor signalling are upregulated in trNK cells. Some of the most highly upregulated genes in trNK cells include *P2ry1* and *P2rx4*, suggesting that purinergic receptor signalling pathway is highly enriched. *Plau*, *Pdgfa*, *Sema6D*, *Bmpr1a*, *Plxna1*, *Itgb3* are all part of smooth muscle cell migration pathway and upregulated in trNK cells (Supplementary Fig. 3F). Other pathways enriched in trNK cells include hormone and sphingolipid catabolism, prostaglandin synthesis, detection of external biotic stimulus, regulation of nitric oxide biosynthesis, regulation of TGF-β production and tissue remodelling. *Msr1*, encoding macrophage-scavenger receptor 1 and *Clec7a*, encoding Dectin-1 exhibited the highest fold change in trNK compared to cNK (about 300-500-fold, respectively). One of the genes upregulated almost 40-fold in trNKs is *Slc40a1*, encoding

ferroportin (FPN1), which is the main exporter of iron from cells, suggesting that iron metabolism may be of importance in uterine trNK cells. Our dataset shows high expression of *Nfe2l2* (NRF2) and significantly upregulated *Nfe2l3* (NRF3) in trNK cells[44]. Both can bind to antioxidant stress response elements and *Nfe2l3* locus has been shown to be associated with endometriosis through genome-wide association studies[45] (Supplementary Fig. 3F).

The fifth comparison between uterine ILC1s and trNK cells highlights transcriptome profiles specific to uterine resident cells (Fig. 3f). The most highly upregulated gene in uterine ILC1s compared to trNK is *Aplnr*, encoding for the adipokine apelin receptor, which is highly expressed in mouse endometrium[46]. Other highly expressed genes in ILC1s include *Adgre4*, encoding the macrophage-specific F4/80 receptor, as well as *Ace*, encoding angiotensin I-converting enzyme of the renin-angiotensin system. The most highly enriched pathway in ILC1s is regulation of protein heterodimerisation activity, including *Hes1*, which interacts with the Notch pathway that is key for ILC development. Among the most highly enriched pathways detected in ILC1s is antigen processing and presentation of peptide via MHC class II, with *H2-Aa*, *H2-DMb1*, *H2-Ab1*, *H2-Eb1* and *CD74* highly expressed, albeit they are also expressed in trNK cells. Enriched in trNK cells and downregulated in ILC1s are *Gzmc*, *Gzmd*, *Gzme*, *Gzmf* and *Gzmg*, assigning tissue-remodelling properties to trNK cells. Most of these granzymes were previously shown to peak in mid- to late gestation[47]. Interestingly, *Xcl1* encoding lymphotactin is highly expressed by trNK cells (Fig. 3f). XCL1 is also produced by human uNK cells and regulates trophoblast invasion[48]. trNK cells highly express *Cd96* encoding for the nectin-binding receptor Tactile, and *Crtam*, encoding a receptor for Nectin-like molecules, *Sh2d1a*, encoding SAP, as well as *Sh2d1b2*, encoding EAT-2B, also a member of SAP-family of adapters. In addition, several T-cell differentiation and activation pathways are enriched in trNK cells, including *CD28*, *Zap70*, and *Itk*. (Fig. 3f). The trNK signature suggests also regulation of myeloid cell differentiation, through *Ets1* and *Itgb3*. Other genes highly expressed in trNK cells regulate protein localisation to plasma membrane, such as *Rab37*, *Sytl2*, *Ptch1*, *Kcnip3*, *Skap1*, *Map7*.

**Uterine g1 ILCs express CXCR6, IL22 and some are ex-ILC3s.** Although our genome-wide transcriptome analysis showed significant upregulation of *Cxcr6* in liver ILC1s (Fig. 3c), both liver and uterine ILC1s express higher CXCR6 protein levels than trNK cells (Fig. 4a). Because there is some overlapping between immature NK cells, ILC1s and ILC3s in other tissues[2], we investigated the lineage relationship among the uterine g1 ILCs using fate mapping reporter mice in which all cells that have expressed, or actively express the RORγt-encoding *Rorc* gene are marked by YFP. Figure 4b shows presence of YFP⁺ cells among uterine g1 ILCs, which are aligned with 'ex-ILC3s' described in other tissues. The human equivalent cells reported previously as stage 3 precursor NK cells[29] or NKp44⁺ ILC3s[15] produce IL-22. We found that trNK in the mouse uterus at mid-gestation also

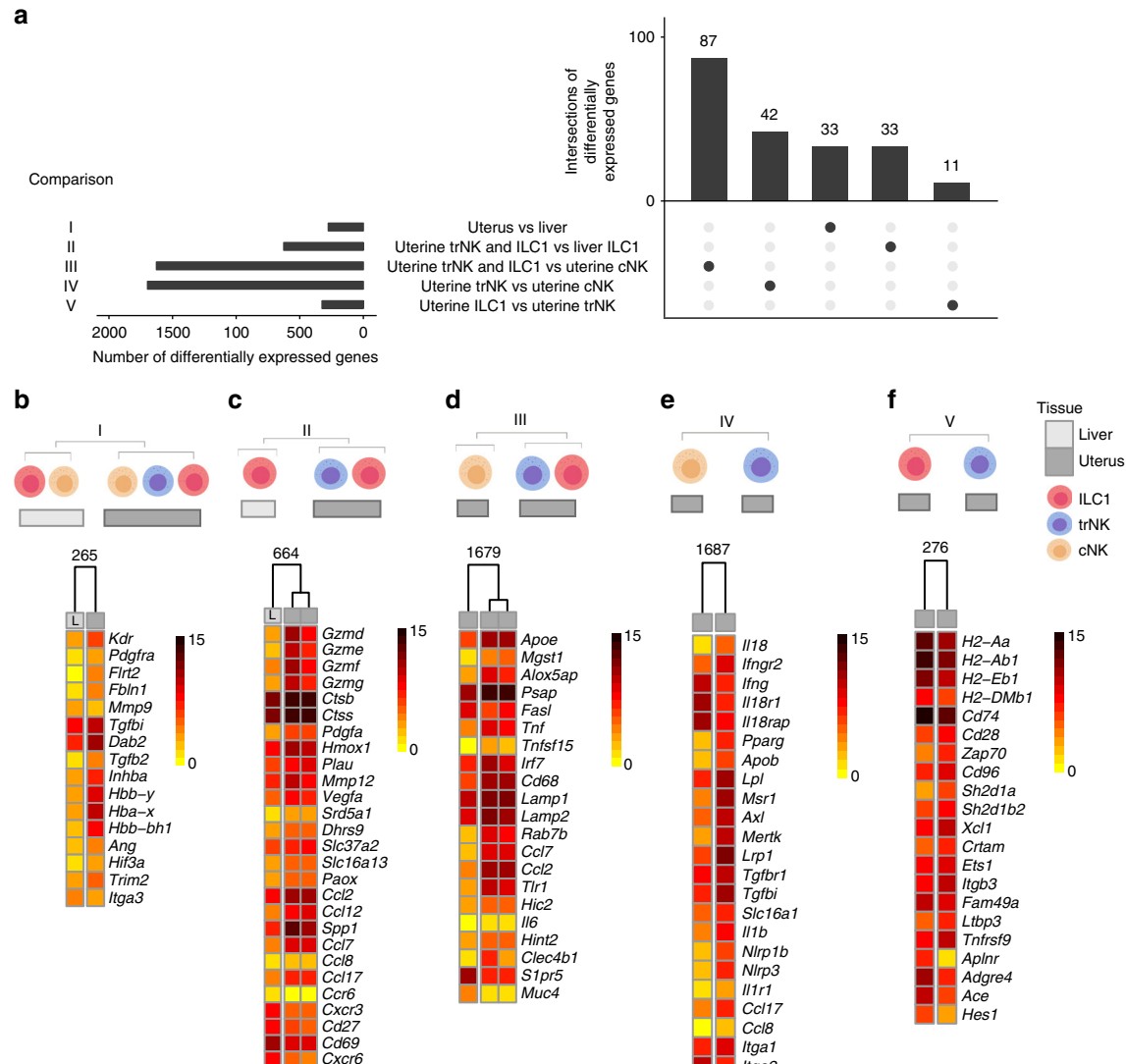

**Fig. 3** Core gene signatures of uterine g1 ILCs. **a** UpSetR plot displaying 5 comparisons of g1 ILCs in uterus and liver. **b–f** Heat maps showing selected genes for comparisons I–V in **a**. Selected genes were chosen from a total list of differentially expressed genes as some representative genes from enriched pathways shown in Supplementary Figure 3A–E as well as from UpSetR lists of intersections of differentially expressed genes. Numbers above heat maps indicate a total number of genes in a corresponding list of differentially expressed genes in that comparison: **b** comparison I: all uterine g1 ILCs vs all liver g1 ILCs. Selected genes were chosen from enriched pathways in Supplementary Figure 3A; **c** comparison II: uterine ILC1s and trNK vs liver ILC1s. Selected genes were chosen from enriched pathways shown in Supplementary Figure 3B; **d** comparison III: uterine ILC1s and trNK vs uterine cNK cells. Selected genes were chosen from enriched pathways shown in Supplementary Figure 3C; **e** comparison IV: uterine trNK vs cNK cells. Selected genes were chosen from enriched pathways shown in Supplementary Figure 3D; **f** comparison V: uterine ILC1s vs trNK cells. Selected genes were chosen from enriched pathways shown in Supplementary Figure 3E

produce IL-22, regardless of the stimulation and, to a lesser extent so can ILC1s (Fig. 4c–e). Although *Cd27* and *Cd69* are both upregulated in liver CD49a$^+$ cells (Fig. 3c), CD27 and CD69 proteins are present at similar levels in both uterine and hepatic cells (Fig. 4f). Both transcripts and Ly49 protein expression levels are slightly higher in trNK cells than in ILC1s and cNK cells within the uterus (Fig. 4f). Despite variable levels of mRNA, all three subsets in the uterus have similar protein levels of NKG2D (Fig. 4f). Bimodal distribution of NKG2A/C/E expression was observed for trNK cells as well. CD62L-encoding *Sell* and CD62L are both upregulated in uterine cNK, similarly to what was reported previously for splenic cNK cells[2]. CD103 is expressed by trNK cells and, to a lower level, by uterine ILC1s, but not by hepatic ILC1s. However, transcripts levels of CD103-encoding *Itgae* are very low in all five subsets, suggesting high mRNA

turnover and potential importance of CD103 in maintaining tissue-residency of g1 ILCs (Fig. 4f). trNK cells were also more proliferative at gd 9.5, as measured by the Ki67 stain (Fig. 4f). A prediction model confirmed unique features of uterine g1 ILCs (Supplementary Fig. 4)

**CXCR6 marks candidate memory uterine g1 ILCs of pregnancy.** Adaptive, or memory NK cells have been described in humans and mice and CXCR6 is associated with memory NK and T cells in mice[49,50]. It is conceivable that subsets of g1 ILCs expand in response to pregnancy-specific cues in second pregnancies and possibly contribute to the lower rate of complications in second pregnancies. We therefore compared the distribution of group 1 ILCs between first and second gestations and found no

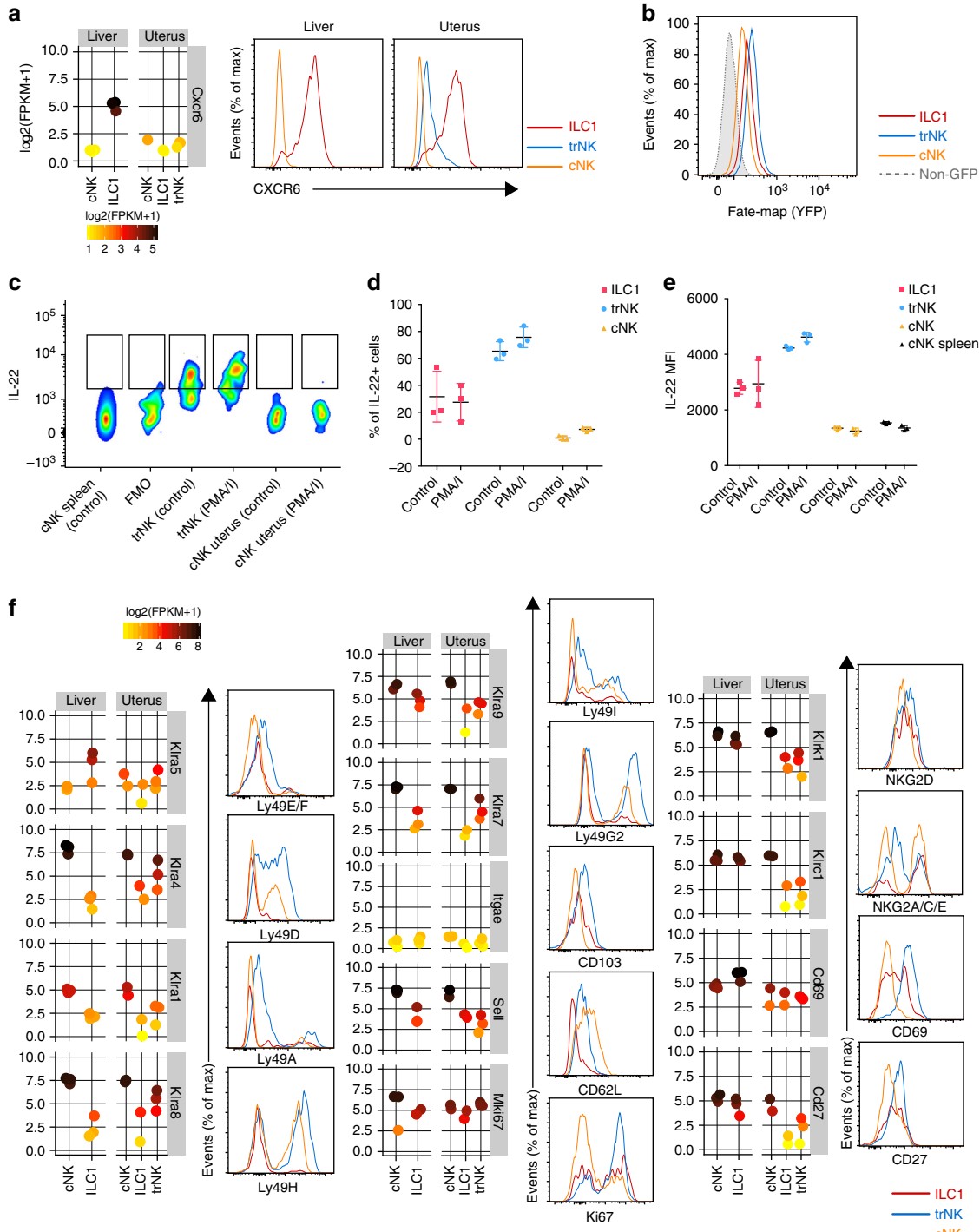

**Fig. 4** Uterine g1 ILCs express CXCR6, IL22 and some cells are ex-ILC3s. **a** Individual gene expression plot showing expression of *Cxcr6* in the liver and uterus. Scale represents log2(FPKM + 1) transformed normalised reads. Histograms are showing CXCR6 protein expression on g1 ILCs in both organs at mid-gestation. Red line-ILC1, blue line-trNK, orange line-cNK. G1 ILCs were gated as described in Fig. 1a. **b** Fate-mapping of g1 ILCs in the uterus of *Rorc(γt)*-*Cre*^TG/R26R females. **c** IL-22 production by uterine trNK and cNK at gd 9.5. Control wells were incubated with medium containing protein transport inhibitors only. **d** Quantification of the percentage of IL-22-producing cells in the uterus at gd 9.5. Error bars represent mean ± SD. **e** Quantification of the mean fluorescence intensity of IL-22 on g1 ILCs at gd 9.5. Error bars represent mean ± SD. **f** Flow cytometry validation of various surface receptors on g1 ILCs in the uterus. Data are representative of three independent experiments. FPKM: fragments per kilobase of exon model per million reads mapped

significant differences in trNK or cNK cells composition (Fig. 5a, b). In sharp contrast, both the frequency and absolute numbers of ILC1s at mid-gestation raise 4–5 fold in second pregnancies (Fig. 5a, b and Supplementary Fig. 5) and ILC1s in second pregnancies upregulate CXCR6 (Fig. 5c), suggesting these cells respond to pregnancy-specific cues and expand in second pregnancies. These results mark uterine CXCR6+ ILC1s as potential memory cells of pregnancy.

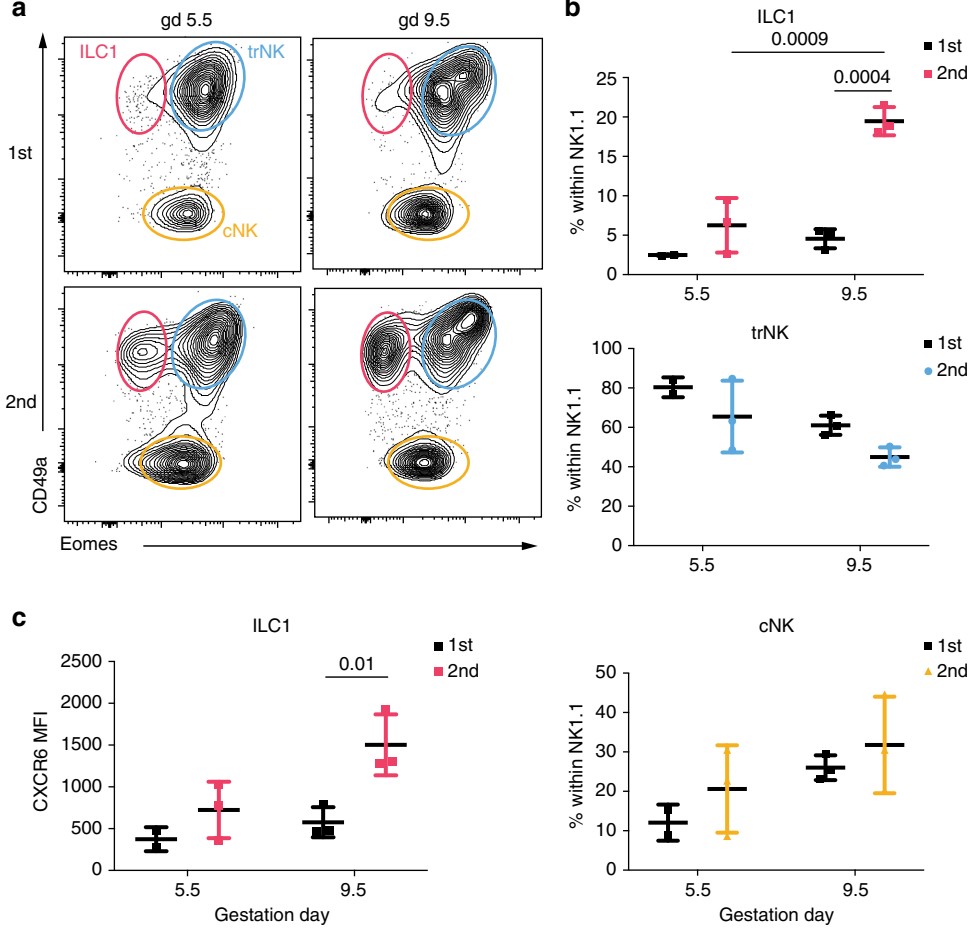

**Fig. 5** Eomes⁻CD49a⁺CXCR6⁺ g1 ILCs expand specifically in second gestation. **a** Representative plots showing distribution of g1 ILCs (gated as in Fig. 1a) at gd 5.5 and gd 9.5 in first (top row) and second (bottom row) pregnancy. Data representative of two or three independent samples and three independent experiments. **b** Quantification of percentages of ILC1, trNK and cNK presented in **a** in first and second pregnancy at gd 5.5 and gd 9.5. Statistical significance was evaluated by two-way ANOVA with multiple comparisons correction. Error bars represent mean ± SD. **c** Mean fluorescence intensity of CXCR6 on ILC1s in first and second pregnancy at gd 5.5 and gd 9.5 Statistical significance was evaluated by two-way ANOVA with multiple comparisons correction. Error bars represent mean±SD

## Disscussion

We show differences in subset composition at different key stages of mouse reproductive life. ILC1s are dominant before puberty, whereas trNKs in early gestation, peaking at gd 5.5 and cNK cells abund in late gestation. The dynamic composition of g1 ILCs, and the whole-genome transcriptome atlas we provide here suggests that g1 ILCs are attuned to uterine remodelling and may have specific functions. The picture emerging is one of trNK cells being the hub of uterine g1 ILCs (Fig. 6), as they interact with the important cell types and factors in the pregnant uterus, including TGF-β, blood vessels, smooth muscle cells, glandular epithelial cells, trophoblast, and leucocytes, including cNK cells. In a cross talk similar to that of macrophages with peripheral NK cells, trNK cells may make IL-18 and stimulate cNK cells in the uterus, which in turn produce IFN-γ and stimulate trNK cells, which unexpectedly also both respond to and produce IL-1. IL-1 and IL-18 are part of the same family of cytokines, which also include the alarmin IL-33, and these are emerging as important factors in reproductive biology and pregnancy complications[12,51]. IFN-γ is the key cytokine required for the reshaping of decidual vasculature in mice that leads to the formation of the placenta. As shown before using histological criteria for subset definitions, cNK cells are the main source of IFN-γ, while other NK-cell

subsets produce angiogenic factors[52]. Our data confirm this division of labour among g1 ILCs, with cNK cells supporting trNK cells to engage with vasculature and other components of the pregnant uterus. Because cNK cells expand after the establishment of the placenta, it is tempting to speculate that they specialise in immune function defending the uteroplacental unit against pathogens.

The core signatures of uterine resident trNK cells and ILC1s are marked by genes involved in tissue homoeostasis, metabolism, genes associated with myeloid cells, cell death, interferon signalling, protein processing and fatty acid biosynthesis. The intra-tissue comparison between uterine trNK cells and cNK cells highlighted genes involved in cholesterol storage, apoptotic cell clearance, aerobic glycolysis, TLR signalling, iron transport, TGF-β signalling as well as *Axl* and *Mertk* of the Tyro-3/Axl/Mer (TAM) family of receptor tyrosine kinases. Gas6, a ligand for Mertk is expressed by trophoblast[53]. TGF-β signalling drives the transdifferentiation of cNK cells into ILC1s, through an intermediate state that is the phenocopy of uterine trNK cells[7]. TGF-β may drive a similar plasticity programme in the uterus whereby cNK and trNK cells convert into ILC1s in second pregnancies, thus partly contributing to generation of candidate memory cells. In our fate mapping experiments plasticity between ex-ILC3s and

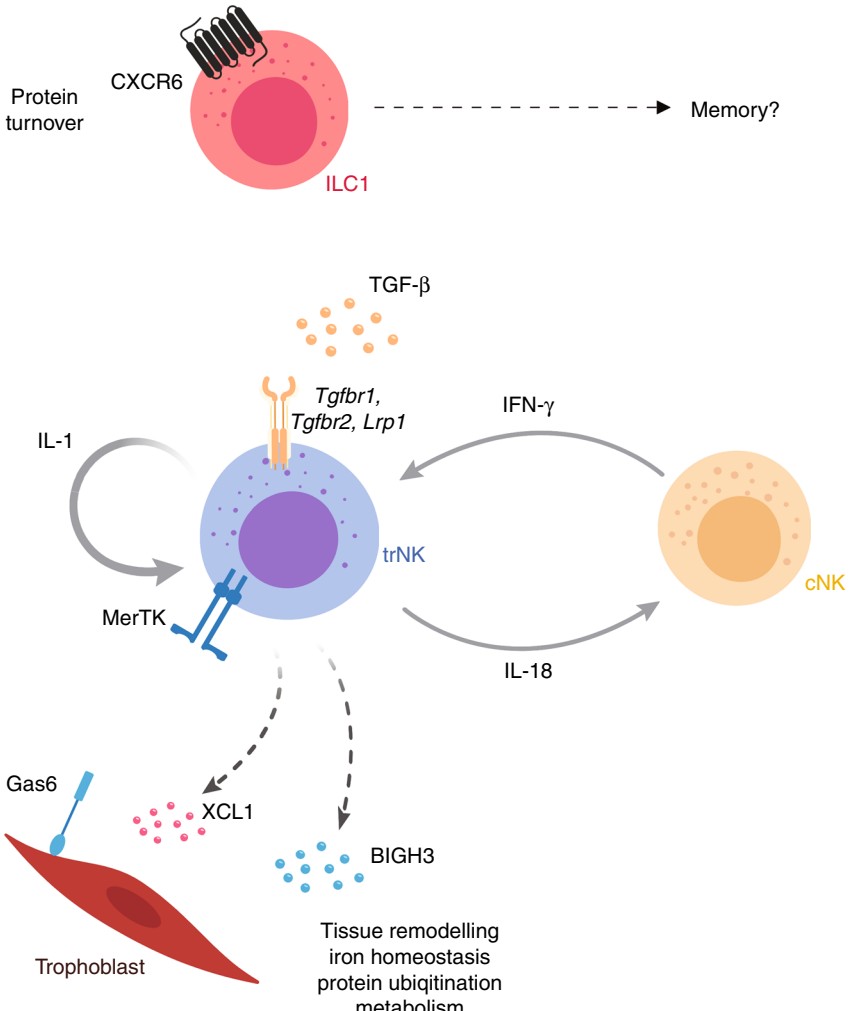

**Fig. 6** Suggested g1 ILC functions in the pregnant mouse uterus. Eomes⁻CD49a⁺ ILC1s are dominant before puberty and specifically expand in second pregnancies, when the expression of CXCR6 is upregulated, marking them as potential memory cells. Eomes⁺CD49a⁺ NK cells (trNK) are most abundant during early pregnancy and may well be the cellular hub of uterine g1 at mid-gestation. They indeed showcase gene signatures of responsiveness to TGF-β, connections with trophoblast, epithelial, endothelial and smooth muscle cells, leucocytes, as well as extracellular matrix. They also express genes involved in anaerobic glycolysis, lipid metabolism, iron transport and protein ubiquitination. Conventional NK cells expand late in gestation and may engage in crosstalk with trNK cells involving IL-18 and IFN-γ

g1 ILCs was evident. Moreover, IL-22 production by mouse trNK cells aligns them with the previously described human stage 3 uterine NK cells[29], also described as NKp44⁺ ILC3s[15].

The comparison between uterine trNK and ILC1s revealed that trNK cells contribute to tissue remodelling through non-cytotoxic granzymes, cell–cell interactions through nectin receptors CD96 and CRTAM, as well as the chemokine XCL-1 and the transforming growth factor-beta induced protein TGFBIp, also known as BIGH3. XCL-1 is expressed by uNK cells, and its receptors by both uterine myeloid cells and invading trophoblast in humans[48]. Therefore XCL-1 emerges as a key molecule in the interactions between maternal lymphocytes and foetal cells, joining other molecular interactions, which include MHC class I receptors NKG2A, human KIR and LILRB, and murine Ly49 (reviewed in ref. [54]). BIGH3 appears as another good candidate for the molecular interactions between resident uterine g1 ILCs and maternal endothelial, smooth muscle cells, as well as foetal trophoblast cells. With trNK and uILC1s expressing various TGF-β receptors TGF-βr1, -βr2 and CD91, it is tempting to speculate that TGF-β induces the secretion of BIGH3 by these cells, which in turn regulates downstream events essential for successful

placentation, such as invasion of trophoblast, vascular remodelling, and angiogenesis. Indeed, previous studies show peak expression of uterine BIGH3 at day 4 of gestation, when trophoblast invasion starts[55].

A number of unexpected genes and pathways were highlighted in trNK cells and ILC1s. Uterine ILC1s may be involved in cross talk to adipocytes through the expression of the receptor for apelin. They also express the angiotensin I-converting enzyme, suggesting a potential role in regulating blood pressure, which is of paramount importance during gestation. Genes involved in complement, IL-1-induced plasminogen activation, as well as haemostasis appear upregulated in trNK cells, which may be involved in inflammation, its accompanying vascular response and the ensuing tissue remodelling. Upregulated in trNK cells are *Fth1* and *Ftl1* encoding ferritin heavy and light chains, with *Fth1* 20-fold upregulated in trNK compared to cNK, and also one of the genes with the highest number of reads. It is therefore possible that trNKs store and export iron or, alternatively, trNK cells respond to oxidative stress induced by iron. Consistently with this possibility, *Hmox1*, which prevents inflammatory tissue injury, is upregulated in trNK cells. Along these lines, genes activated by

oxidative stress and encoding enzymes that detoxify reactive intermediates through glutathione-dependent transferase and peroxidase activities, such as *Mgst1,2* are also upregulated in trNK cells. These pathways are important in the biology of DC and T cells[56,57] and future work will establish if uterine g1 ILCs require these pathways and whether their differentiation is modulated by metabolism. In a recent study, Srebp proteins and genes involved in fatty-acid and cholesterol synthesis appeared essential for the metabolic reprogramming of NK cells in response to cytokine stimulation[58,59]. Interestingly, our analysis reveals that uterine, not liver resident g1 ILCs upregulate *Slc16a13*, which encodes lactate and pyruvate cell exporter MCT13[37]. Warburg-like glycolysis is indeed important for decidual development[60] and trNKs and ILC1s may provide lactate for the development of undifferentiated stromal cells in addition to other sources which function through other MCTs (MCT4). Also, uterine resident g1 ILCs may produce lactate as a signal for arterial remodelling, similar to the tumour micro-environment[61]. The membrane receptor for the high-density lipoprotein cholesterol *Scarb1* is also upregulated in trNK cells, as well as pathways involved in low-density lipoprotein particle remodelling (*Abcg1, Pla2g7, Mpo*) and lipid storage (*Apoe, Soat1, Hexb, Plin2, Gm2a, Ehd1*), suggesting that lipid metabolism may be important for trNK cells. Interestingly, human SCARB1 polymorphism associates with the outcome of in vitro fertilisation[62]. Upregulated in g1 ILCs is also Dectin-1, an important glucan receptor for anti-fungal immunity on macrophages. Recently, peripheral NK cells were shown to recognise beta 1,3 glucan through NKp30[63], suggesting macrophage-like recognition patterns in NK cells too. Consistently with this notion, we detected a number of upregulated genes involved in pathways of microbial molecular pattern recognition. Genes associated with B-cells, or for protein turnover, antigen processing and presentation, as well as the macrophage-associated F4/80 marker are upregulated in ILC1s. The role of these B-cell and myeloid markers on uterine ILC1s is unknown, and a recently described subset of NKB cells is controversial[64,65], but might be induced in SIV and HIV infection[66]. The deconvolution model we have applied here confirms that uterine g1 ILCs have expression profiles aligned with those of cell types other than NK cells and including myeloid cells. A clear limitation of the model is that there are no available ILCs defined in the dataset. Therefore, our results provide new data specific to uterine g1 ILCs at mid-gestation as a new resource for building predictive phenotype models.

Both liver and uterine ILC1s express higher levels of 'memory' marker CXCR6, which prompted us to look for potential changes in the subset composition in first and second pregnancies, with the idea that CXCR6+ ILC1s may be more abundant in second pregnancies. This was indeed the case and we suggest these cells may be associated with memory of pregnancy in the mouse and expand specifically in response to local cues. One possibility is that the CXCR6 ligand CXCL16, expressed also by trophoblast cells, drives ILC1 expansion during pregnancy. Interestingly, pregnancy-trained decidual NK cells have recently been described in humans[67], which may contribute to the lower rate of complications that accompany second pregnancies[34].

In conclusion, we provide a transcriptome atlas of uterine g1 ILCs. Many genes in the UpSet lists are annotated genes that do not have a canonical name yet and include pseudogenes, long non-coding RNA and antisense transcripts. These genes may discriminate cell types and prove instrumental in cell-specific gene targeting. Clearly, there may be heterogeneity even within the subsets we have analysed in the uterus and some of the genes may be specific of smaller populations within the parent population. Future work should capture the dynamic transcriptomic changes within cell types and stages to reveal the biology of uterine ILCs.

## Methods

**Mice.** C57BL/6 (B6) WT mice were purchased from Charles River UK and *Rag2*$^{-/-}$*Il2rg*$^{-/-}$ mice maintained in house. Eomes-GFP reporter mice were a gift of Thierry Walzer[68], *Rorc(γt)-Cre*$^{TG}$/R26R mice from Gérard Eberl[69] and *Rosa26R-EYFP* from Ionel Sandovici. All strains were on B6 background. All animals were used at 8–12 weeks of age and were age-matched for every experiment and all time-matings. The morning of the copulation plug discovery was counted as the gestation day (gd) 0.5 for time-matings. Mice were bred, maintained and mated under pathogen-free conditions at the University of Cambridge Central Biomedical Service in accordance with the University of Cambridge Animal Welfare and Ethical Review Body and United Kingdom Home Office Regulations, as well as at the Animal Facility of the IRCCS-AOU San Martino-IST in accordance with the Italian and European Community guidelines.

**Cell isolation and functional assays.** Mouse uterus, liver and spleen were processed using a protocol involving both mechanical and enzymatic processing. Finely minced tissues were digested in Accutase (Invitrogen) for 35 min at 37°C with gentle orbital agitation (80 rpm)[70] or in Liberase DH (Roche) as described previously[14] for functional assays. Following digestion, tissues were passed through the cell strainer (100 μm for uterus, 70 μm for liver and 40 μm for spleen) using a plunger to mechanically dissociate remaining tissues. Leucocytes were then isolated using 80%/40% Percoll gradient (GE Healthcare Life Sciences) and red blood cell lysis was performed on the resulting cell pellet using BD Pharm Lyse buffer (BD Biosciences) according to manufacturer's instructions. Single cell suspensions obtained in this way were used for downstream analysis and assays. For in vitro stimulation, cells were incubated in a complete medium containing Cell Stimulation Cocktail (plus protein transport inhibitors) during a 4-hour assay, alongside Protein Transport Inhibitor Cocktail for control (both from Invitrogen).

**Flow cytometry.** Following biotin- or fluorochrome-conjugated antibodies specific for the following antigens were used, with the dilutions in the brackets, after the clone: CD45 (clone 30-F11, 100), CD3 (17A2, 50), CD19 (1D3, 100), CD11b (M1/70, 100), NK1.1 (PK136, 100), NKp46 (29A1.4, 50), CD49a (Ha31/8, 300), Eomes (Dan11mag, 100), CD86 (GL1, 50), F4/80 (BM8, 100), B220 (RA3-6B2, 100), MerTK (REA477, 200), CXCR6 (SA051D1, 100), IL-22 (1H8PWSR, 100), NKG2D (CX5), NKG2A/C/E (20d5, 100), CD69 (H1.2F3, 50), CD27 (LG.3A10, 50), CD62L (MEL-14, 100), CD103 (2E7, 25), Ly49H (3D10, 100), Ly49G2 (4D11, 100), Ly49I (YLI-90, 100), Ly49A (YE1/48.10.6, 50), Ly49D (4E5, 50), Ly49E/F (CM4, 100), MHC-II (M5/114.15.2, 400), TruStain fcX (anti-mouse CD16/32, 100) for Fc-receptor blocking and Fixable Viability Dye eFluor 506 (dilution: 1000) for live/dead discrimination. All antibodies were purchased from Biolegend, Invitrogen (eBioscience) or BD Biosciences. For intracellular and intranuclear staining, Foxp3/Transcription Factor Staining Buffer Set (Invitrogen) was used according to manufacturer's instructions. Uterine and liver ILC1 were sorted from Eomes-GFP reporter mice and analysed as live CD45+CD3−CD19−CD11b$^{low/-}$NK1.1+NKp46+CD49a+Eomes−, uterine trNK as live CD45+CD3−CD19−CD11b$^{low/-}$NK1.1+NKp46+CD49a+Eomes+ and uterine and liver cNK as live CD45+CD3−CD19−CD11b$^{low/-}$NK1.1+NKp46+CD49a−Eomes+. Samples were acquired on a LSR Fortessa (BD Biosciences), sorted on FACSAria III instrument (BD Biosciences) and analysed using FlowJo vX software (BD Biosciences).

**RNA-sequencing and analysis.** Group 1 ILCs from uterus and liver were sorted from Eomes-GFP reporter mice into the TRI Reagent (Sigma). RNA extraction was carried out by following TRI Reagent manufacturer's instructions, followed by RNeasy Micro kit (Qiagen). cDNA was amplified using Ovation RNA-seq system V2 (NuGEN) and DNA libraries were produced using Ovation Ultralow System V2 (NuGEN) and following manufacturer's instructions. All mRNA and cDNA quality controls and quantifications were performed using RNA 6000 Pico and High Sensitivity DNA kits (Agilent) on the 2100 Bioanalyzer instrument (Agilent). RNA sequencing was performed on Illumina Hiseq4000 at the Cancer Research UK Cambridge Institute Genomics Core. Data were aligned to GRCm38 mouse genome (Ensembl Release 84) with TopHat2 (v2.1.1, using bowtie2 v2.2.9) with a double map strategy. Alignments and QC were processed using custom Cluster-Flow (v0.5dev) pipelines and assessed using MultiQC (0.9.dev0). Gene quantification was determined with HTSeq-Counts (v0.6.1p1). Additional quality control was performed with feature counts (v 1.5.0-p2), qualimap (v2.2) and preseq (v2.0.0). Differential gene expression was performed with DESeq2 package (v1.16.1, R v3.4.2) and with the same package read counts were normalised on the estimated size factors. Proportions of specific immune cell types from bulk RNA-Seq can be estimated using reference data generated from known proportions of the cell types of interest[71]. DeconRNASeq[72] was applied to take these tables of known cell proportions, defined by gene expression profiles, and used to deconvolute bulk dataset generated in this study to estimate cell proportions within each of the sequence's samples. Gene Ontology (biological process) analysis was performed using PANTHER (Protein ANalysis THrough Evolutionary Relationships database http://www.pantherdb.org). Lists of differentially expressed genes (adjusted $p < 0.05$

and log2 fold change two or above) were submitted to PANTHER Over-representation Test (released 2017-12-05), with Fisher's exact testing utilising a false discovery rate multiple test correction and a *Mus Musculus* reference list. Further details on annotation versions and release dates can be found in the Supplementary Data 1.

**RT-qPCR.** Whole uterus was isolated at time-points of interest, collected in RNA*later* (Invitrogen) to stabilise the RNA and kept at 4 °C overnight. Tissue was homogenized in the Lysing Matrix S homogenisation tubes (MP Biomedicals) on a FastPrep-24 5G Homogenizer (MP Biomedicals) using RNeasy Plus Universal Kit (Qiagen) according to manufacturer's instruction. Isolated RNA quality control and quantification was performed using RNA 6000 Nano kit (Agilent) on the 2100 Bioanalyzer instrument (Agilent). cDNA was amplified using SuperScript VILO cDNA Synthesis kit (Invitrogen) and qPCR was performed using PowerUP SYBR Green Master Mix (Applied Biosystems), on the Quant Studio 6 Flex Real-Time PCR System (Thermo Fisher – Applied Biosystems). *Gapdh* was used as a reference gene. Data were analysed using the ΔΔCt method. Primers were as follows: *Spp1*, 5′-TTCACTCCAATCGTCCCTAC-3′ (forward) and 5′-TTAGACTCACCGCTCTTCAT-3′ (reverse); *Ogn*, 5′-TGCTTTGTGGTCACATGGAT-3′ (forward) and 5′-GAAGCTGCACACAGCACAAT-3′ (reverse); *Ptn*, 5′-TGGAGAATGGCAGTGGAGTGT-3′ (forward) and 5′-GGCGGTATTGAGGTCACATTC-3′ (reverse) and *Gapdh*, 5′-TGCACCACCAACTGCTTAG-3′ (forward) and 5′-GGATGCAGGGATGATGTTC-3′ (reverse). *Ccl2*, *Ccl7* and *Ccl12* were assayed using Thermo Fisher TaqMan (Assays IDs Mm00441242_m1, Mm00443113_m1, Mm01617100_m1) with *Gapdh* as a reference gene (Assay ID Mm99999915_g1).

**Statistical analysis and figure preparation.** Statistical parameters and tests applied are reported in the figure legends. All statistical analyses were performed in Prism 7 (GraphPad Software) with a confidence level of 0.95. *p*-values above 0.05 were considered insignificant and are not indicated in the figures. Figures were prepared using BioRender and Illustrator CC 2018 (Adobe).

## Data availability

All relevant data are available from the corresponding author upon reasonable request. RNA-sequencing data have been deposited in ArrayExpress database at EMBL-EBI under accession number E-MTAB-6812 (https://www.ebi.ac.uk/arrayexpress/experiments/E-MTAB-6812). An open-source repository (https://github.com/CTR-BFX/2018-Filipovic-Colucci) has been created with access to the code used in this study.

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

## Acknowledgements

This work was funded by a Wellcome Trust Investigator Award 200841/Z/16/Z, the Centre for Trophoblast Research (CTR), and the Cambridge NIHR BRC Cell Phenotyping Hub to F.C., the Associazione Italiana Ricerca per la Ricerca sul Cancro (AIRC)-Special Project 5×1000 no. 9962, AIRC IG 2017 Id.19920 and AIRC 2014 Id. 15283 to L. M., and Ministero della Salute RF-2013, GR-2013-02356568 to P.V. I.F. was funded by a CTR PhD fellowship. The Authors are grateful to Thierry Walzer, Gerard Eberl and Ionel Sandovici for sharing resources.

## Author contributions

I.F. designed and performed experiments, analysed the data, wrote the manuscript. L.C. and P.V. designed and performed experiments, analysed the data, and revised manuscript. R.S.H. did the computation analysis of the RNAseq data and edited manuscript. J. M.D. designed and performed experiments, analysed the data and edited manuscript. D. A.H. and T.I. performed experiments. A.S. designed experiments, analysed the data and edited manuscript. M.C.M. and L.M. supervised research and revised manuscript. F.C. designed experiments, analysed the data and wrote the manuscript.

## Additional information

**Competing interests:** The authors declare no competing interests.

