## [Peer Review File · Nature Communications]

Reviewers' comments:

Reviewer #1 (Remarks to the Author):

It is an excellent manuscript of very high quality which provides important new information on the delineation of uterine NK cell subsets. It has important implications for implantation failure, early abortion, intra uterine growth restriction and preeclampsia. I was very much impressed by the overall quality of this work as well as the amount of benchwork it implies. For a manuscript of such quality, I am not keen to make ANY additional comments.

Reviewer #2 (Remarks to the Author):

The manuscript by Filipovic, et al describes a genome-wide transcriptome study of three types of uterine innate lymphoid cells (ILCs) at mid-gestation. For comparison, two analogous innate lymphoid cell types from the liver were also profiled. Comparisons of gene expression were made between different combinations of cell types and differentially expressed genes were identified. Pathway enrichment analysis was performed on these different sets of differentially expressed genes. In addition, cell proportions in the uterus of these three ILCs were measured across different stages of pregnancy and at mid gestation in first and second pregnancies.

I am not an expert in innate lymphoid biology, but have substantial experience in the analysis of transcriptome studies, thus my comments will mainly focus on those aspects of the manuscript and the structure and reporting of the results.

Overall, I have specific concerns with how the clustering analyses were performed, and more generally with the structure of the paper especially concerning the ordering and how results were presented.

1) With respect to the clustering, PCA and hierarchical clustering both seek to determine similarities in features (here expression profiles) from multiple examples (here mice) in an unbiased way. To truly perform an unbiased clustering, the input data should not be filtered based on the knowledge of defined groups (here, different cell types). The hierarchical clustering (and presumably the PCA analysis) was performed using only those genes found to be differentially expressed between some pairwise combination of cell types. Thus, this clustering is no longer unbiased as the data used was based on specific knowledge of a sample's cell type. It is not uncommon to perform filtering of transcription data prior to clustering or PCA analysis, but this should be done in an unbiased way without specific knowledge of sample-specific phenotypes. For instance, it is common to both filter out genes that are lowly expressed across all samples as they are unlikely to contribute to the clustering. It is also common to calculate the variance in expression across all samples and select genes that are highly variable as these are more likely to provide information by which to cluster samples, especially for PCA which specifically looks to explain variance between samples. Without an unbiased clustering, one cannot comment on the similarity or non-similarity of groups of samples based on their positions in a PCA plot or a hierarchical clustering dendrogram.

2) Performing an unbiased clustering would also solve my concern that the "differentially expressed genes" described in the second results section were not defined until the third results section where it was specified that 11 pairwise comparisons were performed. Even so, the full 11 comparisons were never explicitly stated or justified in the text, but rather only listed in Figure 3 with the five that were the subject of further analysis then being listed in the text. It is unclear as to the rationale between why the initial 11 comparisons were performed, and then further why the subset of five were chosen for more in-depth analysis.

3) A good proportion of the paper simply lists differentially expressed genes and enriched pathways based on one of the 5 chosen pairwise comparisons. This results in rather tedious results as it is not clear what is the significance or novelty of these results, does not put these into a broader context, and does not seem to add in a significant way to understanding uterine biology. In general, the full results of differential expression analyses and pathway analyses should be included in supplementary tables. These tables then can be referred to as the full set of results, while general themes or specific results that represent novel and important new knowledge to the field can be highlighted in the text.

4) I find the last section of the results to be the most interesting, but this does not include any information from the expression experiments. I cannot find mention of whether the mice on which the expression experiments were being performed were in their first pregnancy, second pregnancy, or a mix. Since there is a difference in the ILC1s proportions between first and second pregnancy, I would think an interesting question would be whether expression profiles of ILC1s mid-gestation are significantly different.

5) On a more minor note, in Figure 2A, there are only two samples for uterine ILC1s, but in 2B and elsewhere, there are three. It's not clear why this is the case. Also, in Figure 3B-F, the color-coding scheme for the samples along the bottom is not defined. Presumably it is the same as in Figure 2 and other figures, but it should be indicated in all figures.

Response to Reviewer 2

Overall, I have specific concerns with how the clustering analyses were performed, and more generally with the structure of the paper especially concerning the ordering and how results were presented.

Reply: We have swapped the order of PCA (now Fig2A) and heatmap (now Fig 2B) to address the Reviewer's concerns on how results are presented. The text was changed accordingly on pages 8 and 9 and in the Legend for Figure 2.

1) With respect to the clustering, PCA and hierarchical clustering both seek to determine similarities in features (here expression profiles) from multiple examples (here mice) in an unbiased way. To truly perform an unbiased clustering, the input data should not be filtered based on the knowledge of defined groups (here, different cell types). The hierarchical clustering (and presumably the PCA analysis) was performed using only those genes found to be differentially expressed between some pairwise combination of cell types. Thus, this clustering is no longer unbiased as the data used was based on specific knowledge of a sample's cell type.

Reply: We agree that the heatmap is not an unbiased clustering approach and has been generated from the top differentially expressed genes using a log2 fold change cut off of 7.5. The purpose of this heatmap is to visually summarise the data and highlight the proportions of up and down regulated transcripts between the sample groups.

The PCA was produced from the most variably expressed genes from the normalised counts. At no point were the identities of the samples or sets of genes from differential comparisons used to generate the plots. This is therefore a truly unbiased clustering approach.

It is also common to calculate the variance in expression across all samples and select genes that are highly variable as these are more likely to provide information by which to cluster samples, especially for PCA which specifically looks to explain variance between samples. Without an unbiased clustering, one cannot comment on the similarity or non-similarity of groups of samples based on their positions in a PCA plot or a hierarchical clustering dendrogram.

Reply: To explain variance between all samples we show unbiased clustering by selecting the top 250 MV (Most Variable) genes, as suggested by the Reviewer. This is now new Fig 2A. In addition, we also show here below that clustering does not actually change regardless of the number of MV genes selected, i.e. top 250, top 500, 1000 or, as it was shown in the original Fig 2B, with 47,000 MV genes.

2) Performing an unbiased clustering would also solve my concern that the “differentially expressed genes” described in the second results section were not defined until the third results section where it was specified that 11 pairwise comparisons were performed. Even so, the full 11 comparisons were never explicitly stated or justified in the text, but rather only listed in Figure 3 with the five that were the subject of further analysis then being listed in the text. It is unclear as to the rationale between why the initial 11 comparisons were performed, and then further why the subset of five were chosen for more in-depth analysis.

Reply: We do acknowledge that the presentation of data referring to the 11 comparisons might clutter Fig 3A. The choice of the 5 comparisons for more in-depth analysis was informed by the questions we wanted to ask. Namely: what are the differences between ‘circulating’ vs ‘tissue-resident’ subsets within the uterus and how these cells compare to their corresponding subsets in another tissue?

Therefore, for the sake of simplicity, we now cut out the 6 comparisons we do not analyse in the paper (New Fig 3A). The text was changed accordingly on page 9 and in the Legend for Figure 3.

The code allowing data analysis is publicly available nevertheless on GitHub (<https://github.com/CTR-BFX/2018-Filipovic-Colucci>).

3) A good proportion of the paper simply lists differentially expressed genes and enriched pathways based on one of the 5 chosen pairwise comparisons. This results in rather tedious results as it is not clear what is the significance or novelty of these results, does not put these into a broader context, and does not seem to add in a significant way to understanding uterine biology. In general, the full results of differential expression analyses and pathway analyses should be included in supplementary tables.

These tables then can be referred to as the full set of results, while general themes or specific results that represent novel and important new knowledge to the field can be highlighted in the text.

Reply: We have discovered a number of pathways that might help understand how specific lymphocyte subsets contribute to uterine biology – as also stated by Reviewer 1. We do actually distil down our comprehensive dataset by pointing out those pathways in the results section. These pathways are also summarised in Figure 6. The Reviewer is right, however, that we should have provided a Table summarising differentially expressed genes (DEG) and Gene Ontology (GO) Pathways. We do now provide this summary in Supplementary Table 1 (mentioned in the text on page 10) and provide also details of analysis in material and methods on page 6.

4) I find the last section of the results to be the most interesting, but this does not include any information from the expression experiments. I cannot find mention of whether the mice on which the expression experiments were being performed were in their first pregnancy, second pregnancy, or a mix. Since there is a difference in the ILC1s proportions between first and second pregnancy, I would think an interesting question would be whether expression profiles of ILC1s mid-gestation are significantly different.

Reply: All tissues and cells come from first pregnancy, unless specified in the Figure Legends (please see legend to Figure 5 and S5, were we compare first and second pregnancy).

5) On a more minor note, in Figure 2A, there are only two samples for uterine ILC1s, but in 2B and elsewhere, there are three. It's not clear why this is the case.

Reply: We apologise for the mistake. There is indeed a sample missing in the heatmap. The correct version is now New Fig 2B, which presents data with a more stringent P-value cut-off of 0.01 instead of the original 0.05.

Also, in Figure 3B-F, the color-coding scheme for the samples along the bottom is not defined. Presumably it is the same as in Figure 2 and other figures, but it should be indicated in all figures.

Reply: We have included the color-coding scheme for Figure 3B-F and added details of how figures were prepared on page 7.

REVIEWERS' COMMENTS:

Reviewer #2 (Remarks to the Author):

The authors have addressed all of my previous concerns, and I am satisfied that the analyses have been done correctly and are accurately reported.

I just have a couple minor suggestions:

1) In Figure 3A, the first part on the left is fairly self-explanatory (though the bar for comparison 1 seems to indicate more than 265 differentially expressed genes - it is longer than comparison 5 which has 276 diff genes), but I don't understand the second part on the right. What intersections were done? What do the numbers correspond to? The UpSet software that was used to create this is not described anywhere and a reference to a paper on this does seem to be present, and what it does is not described in the methods, results, or figure legend as far as I can tell.

2) For the initial listing of the comparisons performed, just the small addition of descriptive reminders of what the cell types indicated correspond to would help give some initial intuition of these comparisons. For example (though you could definitely improve on this):

- i) All liver g1 ILCs (ILC1 + cNK) vs all uterine g1 ILCs (cNK + trNK + ILC1)
 - ii) Tissue resident liver g1 ILCs (ILC1) vs tissue resident uterine g1 ILCs (trNK + ILC1)
- etc.

This would help remind readers what these specific cell types are.

1) In Figure 3A, the first part on the left is fairly self-explanatory (though the bar for comparison 1 seems to indicate more than 265 differentially expressed genes - it is longer than comparison 5 which has 276 diff genes), but I don't understand the second part on the right. What intersections were done? What do the numbers correspond to? The UpSet software that was used to create this is not described anywhere and a reference to a paper on this does seem to be present, and what it does is not described in the methods, results, or figure legend as far as I can tell.

We thank the Reviewer for pointing this incongruity. This has now been corrected and explained in the text. A revised Figure 3A now displays the correct total numbers as written in 3B-F. The discordance in numbers of genes came from inadvertent use of different software versions of DESeq2 (differences arise from the change in fold change estimators). We have now ensured that the same version of the software was used for all the analyses. We also provide an update to the code repository with options for running the different versions of DESeq using the same fold change estimators. For this reason, we have also corrected Figures 2A.

2) For the initial listing of the comparisons performed, just the small addition of descriptive reminders of what the cell types indicated correspond to would help give some initial intuition of these comparisons. For example (though you could definitely improve on this):

- i) All liver g1 ILCs (ILC1 + cNK) vs all uterine g1 ILCs (cNK + tfNK + ILC1)
 - ii) Tissue resident liver g1 ILCs (ILC1) vs tissue resident uterine g1 ILCs (trNK + ILC1)
- etc.

This would help remind readers what these specific cell types are.

We have done so in the text and also removed the subheading as per Editorial request. So for each comparison we add a short descriptive reminder of what the cell types indicated correspond to.